# Neurodegeneration and Neuro-Regeneration—Alzheimer’s Disease and Stem Cell Therapy

**DOI:** 10.3390/ijms20174272

**Published:** 2019-08-31

**Authors:** Verica Vasic, Kathrin Barth, Mirko H.H. Schmidt

**Affiliations:** 1Institute for Microscopic Anatomy and Neurobiology, University Medical Center of the Johannes Gutenberg University, 55131 Mainz, Germany; 2Institute of Anatomy, Medical Faculty Carl Gustav Carus, Technische Universität Dresden School of Medicine, 01069 Dresden, Germany

**Keywords:** neurodegeneration, Alzheimer’s disease, neuro-regeneration, stem cell therapies

## Abstract

Aging causes many changes in the human body, and is a high risk for various diseases. Dementia, a common age-related disease, is a clinical disorder triggered by neurodegeneration. Brain damage caused by neuronal death leads to cognitive decline, memory loss, learning inabilities and mood changes. Numerous disease conditions may cause dementia; however, the most common one is Alzheimer’s disease (AD), a futile and yet untreatable illness. Adult neurogenesis carries the potential of brain self-repair by an endogenous formation of newly-born neurons in the adult brain; however it also declines with age. Strategies to improve the symptoms of aging and age-related diseases have included different means to stimulate neurogenesis, both pharmacologically and naturally. Finally, the regulatory mechanisms of stem cells neurogenesis or a functional integration of newborn neurons have been explored to provide the basis for grafted stem cell therapy. This review aims to provide an overview of AD pathology of different neural and glial cell types and summarizes current strategies of experimental stem cell treatments and their putative future use in clinical settings.

## 1. Introduction

According to the World Alzheimer Report 2018 there are about 50 million people suffering from dementia in the world. It is expected that this number will increase to about 82 million in 2030, and to about 152 million in 2050. There are over 200 subtypes of dementia, 50%–60% of all cases caused by Alzheimer’s disease (AD). This type of dementia was originally described by Alois Alzheimer in 1907 in Frankfurt am Main, and has become designated later as the most widespread neurodegenerative disease. However, its etiology is still so far unknown, despite many facts about its pathology having been uncovered during the last decades. It is well known that dementia is mainly a sporadic, age-related disease, and only less than 5% of all cases are caused by inherited mutations. The risk of AD development increases 14-fold between the age of 65–85, and affects almost 47% of people over the age of 85 [1]. This disease is characterized by the progressive deterioration of cognitive functions. Clinically, these patients are characterized by an impairment of short-term memory interfering and complicating the activities of daily life, later followed by impairment in the other cognitive fields, e.g., language, logical understanding, orientation, executive function, judgment, behavior and, finally, motor difficulties. This progress is linked to a significant reduction in the volume of the brain in these patients [2]. This atrophy results from the death of neurons and the degeneration of synapses, in particular in the hippocampus formation [3], which is the region responsible for memory and spatial orientation. One of the hallmarks of AD is the occurrence of amyloid plaques in extracellular space of AD brains.

They consist of the agglutinated peptide amyloid-β (mainly Aβ_1–40/42_) containing β-sheet structures and which forms fibrils [4]. The accumulation of amyloid-β peptide is caused by a disturbance of the homeostasis of amyloid-β peptides arising from proteolytic degradation of the amyloid precursor protein (APP). Furthermore, AD brains are characterized by the presence of neurofibrillary tangles, composed of hyperphosphorylated tau protein [5], loss of synapses, dystrophic neurites, and a prominent gliosis.

The pathogenesis of this progressive brain abnormality is multi-factorally caused. The main reasons are age, environmental factors, like chronic stress, traumatic brain injury, internal processes like chronic pain, oxidative stress or inflammation, as well as several genetic factors, like mutations in the genes encoding the amyloid precursor protein (APP), preselinin-1 and -2 or the apolipo-protein E (ApoE) protein.

## 2. Cellular Systems Related to AD

The nervous system is formed mainly by various cell types, neurons and glial cells. Microglia, as well as glial cells, namely oligodendrocytes, astrocytes, and oligodendrocyte progenitor cells (polydendrocytes, NG2 glia), and neurons, may play an important role in the pathogenesis of AD, caused by their roles in neuroprotection, maintenance of CNS homeostasis (concentration of ions, neurotransmitter, etc.), and the brain immune system.

### 2.1. Microglia

Recently, extensive reviews on the role of microglia in healthy brains and the pathogenesis of neurological disorders including AD have been published recently [6,7,8]. Here we will focus upon the main findings, only. Microglia are the brain’s resident macrophages arising from the mesenchyme [9], or are blood-derived (review, see [10]). Microglia display a ramified morphology with numerous branching processes, which enable them to get a survey of the brain environment [11]. Microglial cells have an important function for the preservation of a healthy brain by their ability to attack and remove potential pathogens and detritus, as well as by the secretion of tissue rebuilding factors [12,13]. Furthermore, several data implicate a potential role of microglia for synaptic remodeling [14]. Microglial cells sense neuronal activity, and thereby regulate synaptic plasticity as well as learning and memory mechanisms. Therefore they are a main component in the determination of cognitive function [15,16].

Microglia are linked to neuro-inflammation and are involved in the phagocytic clearance of Aβ peptide. An imbalance in the homeostasis of Aβ peptides as observed in neurological disorders like AD leads to the accumulation and aggregation of Aβ peptide, which is accompanied by a prominent gliosis. This involves changes in the morphology and function of microglia in the Aβ peptide fibrils surrounding parenchyma as an inflammatory response (reviews see [17,18,19,20,21]. The inflammation process is clearly associated with the pathogenesis of the sporadic forms of AD [22]. Microglial cells release pro-inflammatory neurotoxins and cytokine/chemokines like TNF-α, interferon (IFN)−γ or interleukins IL-1β and IL-6.

The human genome-wide association studies (GWAS) have shown that the microglial triggering receptor expressed in myeloid cells 2 (TREM2) plays an important role in the AD-related immune response [23]. It has been shown that TREM2 as a lipid and lipoprotein sensor supports reactive microgliosis [24,25,26] and triggers the transcriptional activation of microglial cells by interacting with the apolipoprotein E4 (APOE4), one of the major risk factors for AD [27].

A further gene which is involved in the etiology of sporadic AD is the ATP-binding cassette subfamily A member 7 (ABCA7) gene, encoding a protein involved in the regulation of lipid homeostasis, and probably also in Aβ homeostasis [28]. This protein indirectly effects the phagocytosis of apoptotic cells and Aβ peptides by mediating the formation of phagocytic cups [29,30,31].

During progress of the neuroinflammatory process in AD pathogenesis, ATP is released into the extracellular space and induces an increased expression of the P2X7 receptor (P2X7R) in microglial cells. This results in the promotion and activation of microglia. P2X7R is an ATP-gated, non-selective cation channel localized in cellular membrane and responsible for the influx of Ca^2+^ or Na^2+^ and efflux of K^+^ [32]. Besides the promotion and activation of microglial cells, an increased expression of this receptor in microglial cells results also in a decrease of their phagocytic activity [33,34,35] and a stimulation of production of neurotoxic molecules like TNF-a, COX-2, IL-6, MMP-9 and reactive oxygen species (ROS) [36,37,38,39]. Hey et al. [40] have shown that P2X7R, modulator of neuroinflammation, plays an important role in mediating microglial cell death and cytokine release, which may be coupled to AKT and ERK pathways. Furthermore, it has been demonstrated that the P2X7/NLRP3 inflammasome pathway leads to the release of the pro-inflammatory cytokines IL-1β, IL-18 and IL-33 [41,42,43]. Therefore, it was suggested that the overactivation of P2X7R in microglial cells mediated by extracellular ATP is most important for the generation of neuroinflammation-induced AD [44].

### 2.2. Glial Cells

#### 2.2.1. Astrocytes

Astrocytes interact with neurons by releasing and recycling glio-transmitter, controlling ion homeostasis, energy metabolism, synaptic remodeling and the modulation of oxidative stress [8]. This cooperative interplay controls neurotransmission as well as synaptic plasticity in different brain regions [45,46,47], and modulates cognitive functions [46,48]. Astrocytes respond quickly to pathological changes inside the brain, react chemotactically to compounds of Aβ plaques, express receptors that bind Aβ, and therefore accumulate at sites of aggregated Aβ depositions, where they take up and degrade Aβ [49,50,51,52]. Extracellular degradation of Aβ may also occur by a secretion of several enzymes like insulin degrading enzyme and metalloproteases [49,53,54,55]. Aggregated Aβ induces an increased Ca^2+^ uptake in astrocytes by an enhanced activation of P2Y_1_ receptors [56], transient receptor potential channel 4 [57], nicotinic acid receptor [58,59] and the glutamate metabotropic receptor mGluR5 [60,61]. Furthermore, uptake of glutamate is decreased by Aβ in astrocytes by changing the activity of adenosine A2A receptors [62].

Most importantly, the presence of Aβ induces the release of cytokines and chemokines by its secretion of Aβ peptides by astrocytes [63]. Apolipoprotein E (ApoE), which plays an important role in AD pathogenesis, is also expressed in astrocytes. Furthermore, hyperphosphorylated tau-forming intracellular tangles, which besides Aβ are the second hallmark of AD, can also be produced in astrocytes (see Section 2.3). Taken together, all these data support the strong connection of astrocytes with the pathogenesis of AD.

#### 2.2.2. Oligodendrocytes

Oligodendrocytes together with myelin lipid layers form the envelope of the neuronal axons required for the fast action potential propagation. They are well known targets for immune reactions in neuronal disorders and exhibit specific morphological changes during AD progression. However, their special function remains still unclear, despite the fact that various experimental data have been published (review see [8]). Oligodendrocytes exhibit specific morphological changes during AD progression. Deterioration in myelin integrity and axonal destruction have been observed [64,65,66,67,68,69]. However, studies on mouse models and hAD samples have shown that there is no reduction of the overall amount of myelin [70,71,72]. Furthermore, it has been demonstrated that Aβ has a cytotoxic effect on oligodendrocytes [73].

#### 2.2.3. NG2-Glia

NG2-glia, also called polydendrocytes or oligodendrocyte precursor cells (OPCs), are the newest discovered glia cell type [74], which play an important role in the pathogenesis of AD. It has been shown that in APP23 mice, Aβ activates GSK3β, resulting in the increased phosphorylation of β-catenin followed by β-catenin degradation accompanied with the inhibition of the Wingles/integrated (Wnt) signaling pathway [75]. This results in an inhibition of the differentiation of NG2-glia [76]. Further studies are required to uncover the specific role of NG-2 glia in the pathogenesis of AD.

### 2.3. Neurons

Neurons express a large number of molecules, protecting them against inflammatory attacks and inducing neurological disorders. Some of these cells have been detected to be injured and to be defective in AD. The most prominent pathological process in neurons is the formation of intracellular neurofibrillary tangles by hyperphosphorylated tau protein, which is one of the pathological hallmarks of AD. The tau protein normally binds to microtubule, and is involved in the axonal transport of mitochondria in the cell. It contains more than 45 phosphorylation sites [77]. The controlled phosphorylation of these sites affects the capacity of tau to bind to microtubules [78]. During AD progress, tau is hyperphosphorylated, dissociates from microtubules and aggregates intracellularly into neurofibrillary tangles [5,79,80,81]. This results in impaired axonal transport of mitochondria between cell body and synapsis, leading to energy dysfunction, generation of reactive oxygen species (ROS) and nitrogen species [82,83].

## 3. Organelles and Organelle Related Processes

### 3.1. Mitochondria

Constantly, mitochondria undergo a balanced fission and fusion process [84] which promotes mitochondrial distribution along micro-tubulare axons into synapses [85]. This process enables mitochondria to react to high energy demands and to facilitate neuroprotective effects, eliminating defective mitochondria constituents and protecting against reactive oxygen species (ROS) damage during ageing [86,87,88].

As mentioned above (see Section 2.3), mitochondria dysfunction has a fundamental role in the pathogenesis of AD [89]. Dysfunctional mitochondria exhibit abnormal morphology-decreased ATP synthesis, impaired antioxidant enzymes and defective oxidative phosphorylation complexes. Mitochondria in the neuronal cells of AD patients show a perinuclear mis-localization resulting in ATP depletion, oxidative stress and synaptic dysfunction [90,91]. The main reason for these disturbances seems to be the aggregation of hyperphosphorylated tau in neurofibrillary tangles (see Section 2.2). However, it needs to be taken into account that this is a very complex process in which also Aβ interactions, ROS production by free metal ions (review see [91], defective autophagy, in particular dysfunctional mitophagy, as well as other processes, may be involved.

### 3.2. Autophagy (Mitophagy)

Autophagy is an organelle-, protein- and lipid-degrading pathway mediated by membranes, vesicles and lysosomes, and is essential for protein, lipid and organelle homeostasis to ensure cell health. Autophagy is the process responsible for the turn-over of mitochondria (called mitophagy) to adapt mitochondria to different energy demands and to eliminate dysfunctional mitochondria. These are actively transported into lysosomes and degraded there. Trafficking may occur via three different pathways:Macroautophagy—engulfment of cytoplasmic components by autophagy vesicles and fusion with lysosomes regulated by autophagy-related proteins and specific autophagy receptors.Microautophagy—direct engulfment of cytoplasmic components by lysosomes [92].Chaperon-mediated autophagy—cytoplasmic proteins are selectively transported into the lysosome [93,94,95,96].

Autophagy is essential for synaptic plasticity, anti-inflammatory function in glial cells, oligodendrocyte development, and the myelination process [97,98]. Recently, it has been demonstrated that autophagy plays a major role in the etiology of AD [99,100,101]. Genetic studies have also demonstrated a link between the expression of several autophagy genes and AD [102,103]. Furthermore it has been shown that impaired mitophagy results in reduced cellular energy levels, increased ROS, and impaired neuroplasticity [104,105]. Recent transcriptome analysis of two AD models (3xTG-AD and Apo3/Apo4-targeted replacement mice) supports the previous data that autophagy, including mitophagy, is clearly dysregulated in AD [106].

### 3.3. Endocytic Processes

Cells use different pathways to internalize materials from the cellular surface into the cell. Most cargos enter cells via clathrin-mediated endocytosis (CME) [107]. Besides these there exist two clathrin-independent endocytotic pathways (CIE), one is using flotillin-1 and -2, and the other one is going via caveolae [108,109]. The cargo including endosomes deliver their material to a variety of intracellular compartments like the endosomes and the trans-Golgi network, or recycle the material back to the plasma membrane [110]. Flotillin-1 and -2 are proteins associated with membrane lipid-rafts, regions enriched with cholesterol, glycosphingolipids and sphingomyelin essential for synapse development, maintenance and stabilization [111,112]. They form micro-domains, and are highly expressed in neurons [113,114]. Caveolae are invaginations in ordered lipid raft domains of the plasma membrane [115], and may contain caveolin-1, -2 or -3 [110,116], depending on cell type. However, all three isoforms are found in the nervous system of mammals [117]. Among them caveolin-1 (Cav-1) is a cholesterol binding protein which organizes and targets synaptic components of the neurotransmitter and neurotropic receptor signaling pathway to the lipid rafts [36,118,119,120,121]. Lipid rafts, cholesterol and Cav-1 may be involved in protection processes against progressing APP and Aβ toxicity. APP is enriched in lipid rafts physically associated with Cav-1 [122]. Overexpression of Cav-1 was associated with α-secretase-mediated proteolysis of APP [117,122]. Cav-2 and Cav-1 interact together and form a heteromeric complex [123]. Contrary to Cav-1, Cav-2 increases with ageing, suggesting a possible increase in the endocytosis of APP [124]. Cav-3 is predominantly expressed in astroglial cells, and is reduced in older cells [124]. Both flotillins are also localized to lipid rafts, and participate in the generation of Aβ [125]. Flotillin-1 is unchanged by ageing, contrary to flotillin-2, which is significantly more highly expressed in aged brains [124].

There are only a few studies on changes in endocytic processes in AD brains, however, remarkable changes in endocytosis have been observed already [108,124,126]. Early endocytic changes like increased volume of total endosomes [127] and increased levels of several CME proteins [128] were detected. Down-regulation of Cav-1 increased the accumulation of APP in AD [122,129]. This is supported by studies of Cav-1 KO mice which develop CNS pathology similar to AD [120,126,130,131,132]. An accumulation of flotillin-1 has been detected in the endosomes of neurons in AD in comparison with healthy patients [125].

An overview representing the involvement of specific cell types and the role of organelles and cell processes in the development of AD is presented in Table 1.

## 4. Stem Cell Therapies

### 4.1. Endogenous Regeneration

Adult hippocampal neurogenesis, the generation of adult-born neurons in the dentate gyrus (DG) from residing stem cells, was broadly analyzed in the context of AD. Firstly, the hippocampus and its neurogenic DG are one of the brain regions that have a key role in learning and memory [133,134,135,136,137]. The hippocampus is one of the first regions of the brain to suffer from damage, observed by the symptoms such as short-term memory loss and disorientation. Secondly, severe neuronal loss in hippocampal regions is another hallmark of AD. Approximately, 1 million neurons are lost in the DG, and about 5 million in the CA1 [138]. On the contrary, the rate of hippocampal neurogenesis was estimated to be about 700 adult-born neurons a day in the adult brain, but is moderately declining with age [139]. Therefore, hippocampal neurogenesis was suspected as the brain’s endogenous regenerative process that may be used for treatment purposes.

Despite the idea that studies of adult neurogenesis in animal models of AD provided a great variability in results, which was dependent on experimental conditions, the age or promoter studied, and though neurogenesis was decreased and increased in its dependence of study conditions (reviewed in [140,141]), most strategies focused on the induction of neurogenesis. Adult neurogenesis can be stimulated through extrinsic administration of chemical agents and growth factors in the context of AD, such as: Erythropoietin [142,143,144], fluoxetine [145], granulocyte colony stimulating factor (G-CSF) and AMD3100 [146], brain-derived neurotrophic factor (BDNF) [147,148,149], insulin growth factor-1 (IGF-1) [150], nerve growth factor (NGF) [151,152], vascular endothelial growth factor (VEGF) [153], growth, and transforming growth factor β (TGF-β) [154]. In addition, neurogenesis becomes stimulated by voluntary running and physical exercise, as well as an enriched environment [155,156,157,158]. Long-term physical activity was even found to diminish neuronal loss in the CA1 region in the AD transgenic mouse model [159], which is of particular interest.

The replacement of CA1 neurons is still unclear, as this is one of the major destruction sites. Therefore, further and additional approaches are required.

### 4.2. Engrafted Regeneration

In general, there are several major groups of stem cells that are used for therapeutic purposes. Mostly, embryonic stem cells (ESCs), neural stem cells (NSCs), induced pluripotent stem cells (iPSCs) and mesenchymal (MSCs) stem cells are applied in this context. Different strategies regarding their application in the context of stem cells therapies in AD is overviewed in Figure 1. Other types of stem cells, such as olfactory ensheathing cells and hematopoietic stem cells, may as well be applied for treatment [160], however they are not discussed in this review.

#### 4.1.1. ESCs

ESCs possess enormous potential due to their pluripotency, the ability to generate cell types from the ectodermal, mesodermal and endodermal germ layers, and due to their unlimited self-renewal capacities [161]. This makes them a very attractive candidate as a therapeutic means. They are derived from the inner cell mass of a developing blastocyst at embryonic day 5 to 6. For research purposes however, most ESCs are derived from embryos that develop from eggs that have been fertilized in vitro and then donated for research purposes (with the informed consent of the donors). Nevertheless, the use of ESCs for research purposes naturally remains a controversy, and is differentially regulated from country to country. For example, in Germany the use of ESCs is prohibited, and yet in Austria it is allowed. In the USA, on the federal level, there is no official ban, however particular states employ their own restrictions.

Several reports have explored the role of ESCs in AD rodent models. Pluripotency, one the biggest advantages of ESCs, represents one of their main drawbacks, as their differentiation can occur towards any direction, and lead to neoplasia or teratoms [162,163]. Therefore, research strategies focus on establishing pre-differentiation protocols. Mouse ESCs (mESCs) were successfully used to generate basal forebrain cholinergic neurons (BFCNs), which are severely affected in AD patients, and this process was applied for driving ESC-derived, as well as primed NPC differentiation, upon the transplantation into the AD rat model [164]. In addition, these rats displayed significant behavioral improvement in memory deficits. Human ESCs (hESCs) were also able to produce cholinergic neurons in vitro and upon engraftment into cultured entorhinal-hippocampal mouse slices, they connected to the existing neuronal network [165]. Similarly, both mESCs and hESCs were directed into mature BFCNs, and following transplantation into AD mice, improvement in learning and memory performance was observed [166]. Another approach was to differentiate hESCs into MGE (medial ganglionic eminence)-like progenitors cells, as MGE is the place of origin of basal forebrain neurons, including BFCNs and GABA interneurons, during development. Transplantation of these MGE-like progenitors into the mouse hippocampus yielded comparable results as the studies mentioned above [167].

#### 4.1.2. NSCs

Adult NSCs reside in the sub-granular zone (SGZ) of hippocampal DG and in the subventricular zone (SVZ) of the lateral walls of the ventricles. They are self-renewing, multipotent cells that can give rise to neurons, oligodendrocytes or astrocytes [168]. Discovery of the NSCs’ existence in the adult brain a few decades ago opened a whole new field of treatment possibilities for incurable brain diseases. Aside from strategies for their endogenous repair applications, they are also used in exogenous repair strategies. Critical points of NSCs’ transplantation include their poor survival outcomes [169,170], therefore, aside from substituting damaged cells via the differentiation and autocrine production of neurotrophic and neuroprotective factors, therapeutic approaches have focused upon a combination of grafting and the application of beneficial factors and therapeutic genes.

hNSCs from fetal telencephalon were transplanted into the lateral ventricles of an AD mouse brain where they migrated, engrafted and differentiated into neuronal and glial cells. The results of this process included improved spatial memory, decreased tau phosphorylation and Aβ42 levels, reduced microgliosis and astrogliosis [171], enhanced endogenous synaptogenesis [172] and increased neuronal, synaptic and nerve fiber density [173]. Interestingly, these effects were achieved via multiple mechanisms, including the modulation of signaling pathways, metabolic activity, secretion of anti-inflammatory factors and cell-to-cell contacts. BDNF, an important NSC-derived neuroprotective factor, was found to be crucial for improving the cognition of AD rodents with transplanted NSCs (obtained from the brain or the hippocampus), by improving hippocampal synaptic density [147] and enhancing the number of cholinergic neurons [174,175]. Transplantation of an hNSC line that over-expresses choline acetyltransferase, an enzyme that synthesizes acetylcholine (ACh) into the aged ICR mice, led to the improved cognitive function and physical activity of aging mice [176]. This was achieved by producing ACh directly and restoring cholinergic neuronal integrity, probably mediated by increased levels of BDNF and NGF neurotrophins. In addition, hNSCs were genetically modified to express NGF, and were transplanted in mice with induced cognitive dysfunction where they improved learning and memory [152]. Other neurotrophic effects were accomplished by a combination of NSC grafting and the supply of neurotrophic drugs like (+)-phenserine (amyloid modulator) [177] or Cerebrolysin (mixture of neurotrophic peptides) [178], to support the survival of NSCs. Finally, NSCs’ transplantation reduced neuroinflammation by reducing glial and toll-like receptor 4 (TLR4) activation and its downstream signaling pathways [179].

#### 4.1.3. iPSCs

Using defined reprogramming factors to reprogram fully differentiated somatic cells into iPSCs has become a novel strategy to produce pluripotent cells derived from patients that enable autologous transplantation [180]. Although many ethical issues regarding immune rejection can be circumvented by the use of iPSCs, their application, however, bares limitations in terms of AD-related pathological phenotypes of generated neurons (such as abnormal Aβ levels and increased tau phosphorylation) [181,182]. Currently, there is a limited amount of studies using transplanted iPSCs, due to its novelty and mentioned issues, however their application in vitro might be the best tool for studying AD pathology and screening for therapeutic drugs.

Under in vitro conditions, hiPSCs derived from skin fibroblasts of patients (with APOE alterations) were used to produce neurons that expressed apolipoprotein E4 (ApoE4) variant, which is a major risk factor for AD. The study identified ApoE4 and its altered conformation (and not ApoE3, which showed rescue properties) as the cause of the pathogenic phenotype of AD, and provided the possibility of treatment by applying a corrector of the pathogenic conformation of ApoE4 [183]. Transplantation of iPSCs in vivo was achieved with iPSCs derived from mouse skin fibroblasts by treating protein extracts of embryonic stem cells. These cells differentiated into glial cells and decreased plaque depositions in the 5XFAD transgenic AD mouse model and moreover, improved cognitive dysfunction [184]. Separately, neuronal precursors developed from human iPSCs were transplanted into the hippocampus of transgenic mice with severe amyloid β deposition and progressive spatial memory dysfunction, where they differentiated into cholinergic neurons and significantly improved memory impairments [185]. In addition, iPSCs transplantation was successfully used in various Parkinson’s disease models [186,187,188] and stroke models [189].

#### 4.1.4. MSCs

Mesenchymal stem cells are multipotent stromal cells that can be isolated from various organs and tissues, such as bone marrow (BM-MSCs), umbilical cord (UCB-MSCs), adipose tissue (AD-MSCs), peripheral blood, amniotic fluid, muscle and lung [190]. MSCs can differentiate into a variety of cells types, are highly proliferative and are easily accessed and easily handled. Therefore, these cells have found a very broad application in research. However, the rates of neuronal differentiation tend to be rather low, and glial cell generation can be preferred [191]. Nevertheless, MSCs can also be applied intravenously, as these cells are able to cross the blood-brain barrier and travel to the injury site [192,193], which offers enormous benefits for patient treatments.

Similarly to other types of stem cell transplantation effects, transplantation of MSCs reduced Aβ deposits and tau-phosphorylation, increased neurogenesis and provided support with secreted factors and improved learning and memory deficits [194,195,196,197,198]. Immune modulating and anti-inflammatory effects have also been observed, through the upregulation of neuroprotective and downregulation of pro-inflammatory cytokines. Another important pathway by which MSCs participate in tissue repair is by the secretion of extracellular vesicles (EVs), particularly exosomes and micro-vesicles (MV), and this approach is extensively explored. MSCs can be genetically altered to release EVs supplemented with therapeutic agents, including siRNAs and enzymes, that target Aβ deposits [199,200]. Alternatively, MSCs can be manipulated to overexpress cytokines like VEGF that exhibit regenerative effects in AD models [201].

### 4.3. Translation

There is an increasing number of clinical trials that employ stem cell therapies in order to treat Alzheimer’s disease, and most of them are still ongoing. The treatment concept includes the capability of transplanted stem cells to differentiate into neuronal and glial cells that suffered the damage, to act in a paracrine manner by secreting neurotrophic and neuroprotective agents and to stimulate endogenous repair mechanisms. The most-used cell type for this purpose is the MSC, due to easy harvest, the possibility for intravenous transplantation, lack of immune reaction and ethical issues. The advantages and disadvantages of distinct stem cell types are presented in Table 2.

In one of the first studies (ClinicalTrials.gov Identifier: NCT01297218) allogenic hUBC-MSCs were injected intracranially into bilateral hippocampi of AD patients in phase I mostly to test safety and cells dosage, followed by effect evaluation phase (ClinicalTrials.gov Identifier: NCT01696591) where no Aβ decrease or cognitive improvement was observed, but limitations of the study need to be considered [202]. Another study in a phase I clinical trial focused on employing autologous fibroblasts transduced to express nerve growth factor (NGF) and could show positive evidence of a response to NGF through improved function of cholinergic neurons followed by improved cognitive decline after 22 months [203]. In a larger clinical trial (25 subjects) BM-MSCs from healthy donors are used intravenously subjects will be followed up at 2,4,13, 26, 39 and 52 week post study product infusion with expected end on March 2020 (ClinicalTrials.gov Identifier: NCT02600130). Similarly, BM-MSCs are used intravenously but in combination with near infrared light which should provide additive support (ClinicalTrials.gov Identifier: NCT03724136) on 100 participants with an estimated completion date on October 2021. In another study autologous AdMSC will be applied (ClinicalTrials.gov Identifier: NCT03117738) where intravenous administration will be repeated 9 times at a 2-week interval.

## 5. Conclusions

Since Aloysius Alzheimer’s first publication on a patient exhibiting a wide-spread case of dementia, it became more and more evident that the etiology of this disease is a multifactorial-caused process characterized by a high neuropathological heterogeneity. In spite of an explosion of studies on the neurophysiology of dementia, we are still far from a complete understanding of the etiology of AD, and a more global view has to be developed. Nevertheless, some new data may be a basis for developing new concepts for the successful treatment of AD.

Stem cell therapy bares enormous potential for the treatment of AD. Pre-clinical analyses yielded successful results and paved way for clinical trials; however, it is evident that translation from rodent models to AD patients is not straightforward. Issues with participant enrollment, choosing the right time for transplantation, possible issues regarding gender differences and long duration of monitoring are all factors that affect the outcomes of clinical trials. Therefore, a comprehensive evaluation of the key factors, including the cell type and source, delivery systems, long-term safety and efficacy, the reaction of the implanted cell to the AD environment and mechanisms of action in the AD model, is needed before prior to the start of clinical trials. Altogether, it takes rather a long time before conclusions can be drawn and the results of currently ongoing studies will provide answers on the efficiency of stem cell therapy in AD patients. AD is a progressive disease which is initiated usually years before the onset of first symptoms. To this day, early diagnosis remains one of the most crucial points in AD treatment. In this context, additional research is needed that can provide reliable diagnostic tools. Moreover, it is possible that a synergy of methods needs to be employed in treatment strategies which would involve exogenous neuroreplacement, endogenous neurogenesis, genetic manipulations and pharmacological agents.

## Figures and Tables

**Figure 1 ijms-20-04272-f001:**
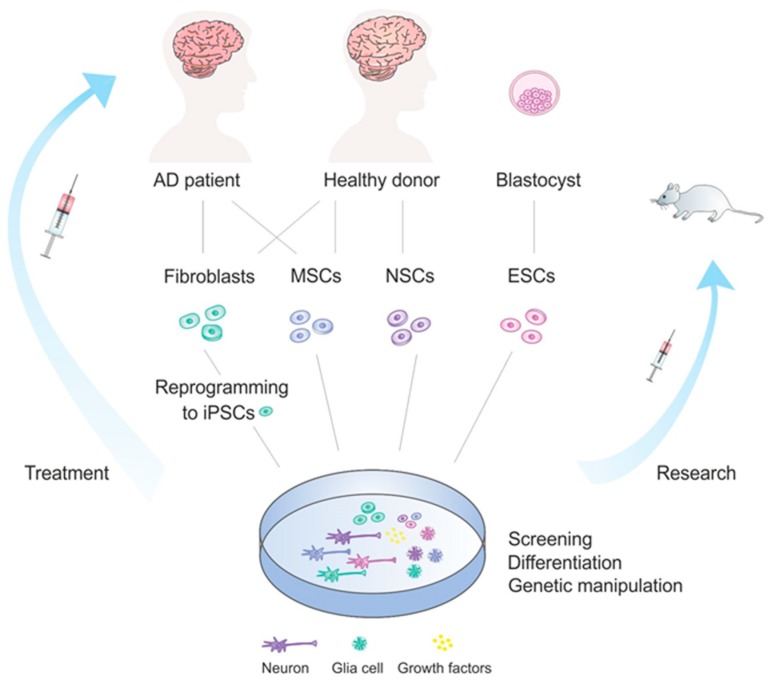
Stem cell treatment development in Alzheimer’s disease. AD patients or healthy donors, or a blastocyst in case of ESCs, can be the source of different stem cell types which can be used for treatment development. Stem cells can be further involved in various processes, such as differentiation, genetic manipulation, used for drug screening or tested on mice and other organisms. Finally, stem cells reach the AD patients and are applied in therapeutic purposes.

**Table 1 ijms-20-04272-t001:** Involvement of specific cell types and the role of organelles/cell processes in the development of AD.

Cellular System	Physiological Function	Involvement in AD
Microglia	Component in the determination of cognitive function, preservation of a healthy brain, attack and removal of pathogens and detritus, secretion of tissue rebuilding factors, synaptic remodeling [15,16,17,18,19,20,21,22,24,25,26,27]	Involved in the generation of neuro-inflammation, imbalance of Aβ peptide homeostasis, decrease of phagocytic activity, release of pro-inflammatory neurotoxins and cytokines/chemokines [29,30,31,33,34,35,36,37,38,39,40,41,42,43,44]
Astrocytes	Interactions with neurons by releasing and recycling glio-transmitter, control ion homeostasis, energy metabolism, synaptic remodeling and the modulation of oxidative stress leading to control of neurotransmission, synaptic plasticity and the modulation of cognitive functions involved in the degradation of Aβ peptides [8]	Changes in intra- and extracellular degradation of Aβ peptides, release of cytokines and chemokines, expression of ApoE, formation of hyperphosphorylated tau protein [45,46,47,48,49,50,51,52,53,54,55,56,57,58,59,60,61,62,63]
Oligodendrocytes	Form together with myelin lipid layers the envelope of the neuronal axons [8]	Specific morphological changes during AD progression, deterioration in myelin integrity and axonal destruction, killed by Aβ peptides [64,65,66,67,68,69,73]
NG2-glia	Oligodendrocyte precursor cells	Aβ peptides-induced inhibition of Wnt signaling pathway results in an inhibition of the differentiation of NG2-glia [74,76]
Neurons	Expression of a large number of molecules for protection against inflammatory attacks and induction of neurological disorders [5]	Formation of intracellular neurofibrillary tangles by hyperphosphorylated tau protein, impaired axonal transport of mitochondria resulting in energy dysfunction, generation of reactive oxygen and nitrogen species [77,78,79,80,81,82,83]
Mitochondria	ATP synthesis, reaction to different energy demands by balanced fission and fusion processes and directed transport along axons, protection against ROS damage by elimination of defective constituents	Aggregation of hyperphosphorylated tau in neurofibrillary tangles, perinuclear mis-localization resulting in ATP depletion, synaptic dysfunction, oxidative stress [84,85,86,87,88,89,90,91]
Autophagy (Mitophagy)	Degradation of organelles, proteins and lipids mediated by membranes, vesicles and lysosomes, essential for organelle turn-over, synaptic plasticity, anti-inflammatory function in glial cells, oligodendrocyte development, and the myelination process	Dysregulation leading to changes in the expression of several autophagy genes resulting in reduced energy levels, increased ROS production and impaired neuroplasticity [92,93,94,95,96,97,98,99,100,101,102,103,104,105,106]
Endocytic processes	Internalization of materials from the cell surface by clathrin-dependent and clathrin-independent pathways, using flotillins or caveolins as the main proteins, as well as the protection against the processing of APP and Aβ toxicity [36]	Only a few data available, volume of total endosomes increases, enhanced levels of several endocytic enzymes [107,108,109,110,111,112,113,114,115,116,117,118,119,120,121,122,123,124,125]

**Table 2 ijms-20-04272-t002:** Summary of advantages and disadvantages of distinct stem cell types.

Stem Cells	Advantages	Disadvantages
ESCs	Unlimited self-renewal; Pluripotent [161]	Ethical issues; Uncontrolled differentiation [162,163]
NSCs	Multipotent [168]	Poor survival [169,170]
iPSCs	Autologous transplantation; Pluripotent [180]	Possible pathological phenotype [181,182]
MSCs	Easy handling; Multipotent; Intravenous application [190,192,193]	Low rate of neuronal differentiation [191]

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
