# Peer review of "Neurodegeneration and Neuro-Regeneration—Alzheimer’s Disease and Stem Cell Therapy"

_ijms, 2019, doi:10.3390/ijms20174272_

Round 1

Reviewer 1 Report

The authors summarized the cellular systems related to Alzheimer’s disease (AD) and stem cell therapy against AD. It’s well-written like an encyclopedia, and enough for the acceptance with some minor remarks.

In the title, “stem cell therapy” should be included, otherwise the authors have to review broad spectrum of AD therapies including pharmaco- and gene-therapies. To make it easier for readers to understand, table for advantages and disadvantages (feasibilities s and limitation) of all stem cell therapies should be prepared. I think “Translation” is one of the most attractive sections for readers. So, I’m happy to see the table in this section.

Author Response

We thank the reviewer for the comments and the opportunity to significantly improve our manuscript. In this revision, we have addressed the concerns of the reviewer thoroughly. All changes in the text are indicated in red in the revised manuscript and also depicted in this point to point response. Please find below our responses to the reviewer’s comments.

Point 1: In the title, “stem cell therapy” should be included, otherwise the authors have to review broad spectrum of AD therapies including pharmaco- and gene-therapies.

 Response 1: We thank the reviewer for the comment. Indeed, the observation is fully correct and therefore the title of the manuscript has been edited to ‘Neurodegeneration and Neuroregeneration - Alzheimer’s disease and Stem cell therapy’.

Point 2: To make it easier for readers to understand, table for advantages and disadvantages (feasibilities s and limitation) of all stem cell therapies should be prepared. I think “Translation” is one of the most attractive sections for readers. So, I’m happy to see the table in this section.

Response 2: We thank the reviewer for the comment. Based on your suggestions a table (Table 2) was incorporated into the section ‘Translation’ which summarizes advantages and disadvantages of stem cell therapies in AD. We hope this provides better understanding and overview of different stem cell therapies.

Reviewer 2 Report

This review article is overall interesting and of good quality. The focus on stem cell therapy is especially timely and the literature cited on this topic is exhaustive and recent.

My main concern on the article is lack of clear illustrations. Figure 1 contains conceptual errors (e.g., "ESCs" and "reprogramming" are inappropriately used). In any case, as it is the figure adds little to the whole article and in fact it is not even mentioned in the text. It would have been much more useful if the Authors had provided Tables synthesising specific literature data, or figures showing the involvement of specific cell types and/or the role of specific organelles/cell processes.

Author Response

We thank the reviewer for the comments and the opportunity to significantly improve our manuscript. In this revision, we have addressed the concerns of the reviewer thoroughly. All changes in the text are indicated in red in the revised manuscript and also depicted in this point to point response. Please find below our responses to the reviewer’s comments.

Point 1: My main concern on the article is lack of clear illustrations. Figure 1 contains conceptual errors (e.g., "ESCs" and "reprogramming" are inappropriately used). In any case, as it is the figure adds little to the whole article and in fact it is not even mentioned in the text.

 Response 1: We thank the reviewer for the comment. We agree that Figure 1 didn’t provide an overview that was precise enough. Therefore, Figure 1 was edited and we hope that now it provides a clear overview of the stem cell therapies discussed in the manuscript and contributes to better understanding of the text. It is also appropriately mentioned in the text (line 254).

Point 2: It would have been much more useful if the Authors had provided Tables synthesising specific literature data, or figures showing the involvement of specific cell types and/or the role of specific organelles/cell processes.

Response 2: We thank the reviewer for the comment. Based on your suggestions a table (Table 1) was incorporated into the manuscript which summarizes the involvement of specific cell types and the role of organelles and cell processes in the development of AD. We hope this provides better understanding and overview of the manuscript.